# Mining of Thousands of Prokaryotic Genomes Reveals High Abundance of Prophages with a Strictly Narrow Host Range

Gamaliel López-Leal,[a,*] Laura Carolina Camelo-Valera,[a] Juan Manuel Hurtado-Ramírez,[c] Jérôme Verleyen,[c] Santiago Castillo-Ramírez,[b] Alejandro Reyes-Muñoz[a]

[a]Grupo de Biología Computacional y Ecología Microbiana, Max Planck Tandem Group in Computational Biology, Departamento de Ciencias Biológicas, Universidad de los Andes, Bogotá, Colombia
[b]Programa de Genómica Evolutiva, Centro de Ciencias Genómicas, Universidad Nacional Autónoma de México, Cuernavaca, Morelos, México
[c]Instituto de Biotecnología, Universidad Nacional Autónoma de México, Cuernavaca, Morelos, México

**ABSTRACT** Phages and prophages are one of the principal modulators of microbial populations. However, much of their diversity is still poorly understood. Here, we extracted 33,624 prophages from 13,713 complete prokaryotic genomes to explore the prophage diversity and their relationships with their host. Our results reveal that prophages were present in 75% of the genomes studied. In addition, Enterobacterales were significantly enriched in prophages. We also found that pathogens are a significant reservoir of prophages. Finally, we determined that the prophage relatedness and the range of genomic hosts were delimited by the evolutionary relationships of their hosts. On a broader level, we got insights into the prophage population, identified in thousands of publicly available prokaryotic genomes, by comparing the prophage distribution and relatedness between them and their hosts.

**IMPORTANCE** Phages and prophages play an essential role in controlling their host populations either by modulating the host abundance or providing them with genes that benefit the host. The constant growth in next-generation sequencing technology has caused the development of powerful computational tools to identify phages and prophages with high precision. Making it possible to explore the prophage populations integrated into host genomes on a large scale. However, it is still a new and under-explored area, and efforts are still required to identify prophage populations to understand their dynamics with their hosts.

**KEYWORDS** prophages, viral diversity, bacteria, archaea, bacteriophages, eubacteria

Prokaryotic viruses (phages) are considered the most abundant biological entities on the planet (1). These phages reproduce either through a lytic cycle, like virulent phages, or vertically by integrating into the host genome and taking advantage of its replication cycle as prophages (2). This integration into the host chromosome can alter the host phenotype by disruption of open reading frames and by altering the expression of flanking genes. Furthermore, this process contributes as a major source of new genes and functions within the bacterial genome (3, 4). This gene turnover contributes to the fitness and the appearance of ecologically important bacterial traits such as virulence factors, drug resistance mechanisms (5–7) or phage-derived bacteriocins and tailocins (8, 9). In this regard, there are some reports that prophages are more frequently present in pathogenic than in nonpathogenic strains (10). For example, the comparison genomes of laboratory strains from *Escherichia coli* and their pathogenic counterparts revealed that the main differences between these strains were due to the insertion of prophages or other genetic mobile elements (11–13). Moreover, pathogenicity islands, commonly acquired by horizontal gene transfer are a general mechanism by which many bacteria display a pathogenic phenotype (12). These observations suggest that

Address correspondence to Gamaliel López-Leal, gamlopez@ccg.unam.mx.

*Present address: Gamaliel López-Leal, Grupo de Genómica y Dinámica Evolutiva de Microorganismos Emergentes. Consejo Nacional de Ciencia y Tecnología, Ciudad de México, México.

The authors declare no conflict of interest.

10.1128/msystems.00326-22   **1**

the prophages play an important role in the evolution of their hosts (14). However, these findings come from studies focusing on very particular species.

Until not long ago, the identification of prophage regions had been computationally challenging due to the lack of information about the diversity of phage sequences. However, recently, along with the development of sequencing technologies, the expansion of phage-derived sequence databases has been increasing and, with it, powerful tools for studying phages and prophages (15). These recent developments have allowed to efficiently identify prophage regions from prokaryotic genomes (16). Moreover, the advance in sequencing technologies has produced dozens to hundreds of high quality bacterial and archaeal genomes (17, 18), and with this, we have unintentionally sequenced thousands of prophages. Since prophages are part of the host genome, they are *in situ* recovery from their host providing vast information on their diversity and on the relationship with their host. In this sense, few studies have explored the diversity of prophages from genomes and publicly available metagenomic data for elucidating prophage-bacteria relationships (19–21). However, those studies have analyzed modest numbers of bacterial genomes and have focused mainly on very particular aspects, such as the horizontal gene transfer (20), the relationships of the prophage-CRISPR-Cas systems, the contribution of host genome size and prophage acquisition and the dominance of commensal lysogens in a particular niche (19). Nonetheless, not much attention has been paid to other variables such as (i) the phylogenetic relationships of the host and the abundance of the prophages, (ii) the presence of prophages in pathogens, and (iii) the host range of prophages and their relatedness in genetic repertoires. Therefore, to analyze these variables and to obtain insights into the knowledge associated with the prophage diversity and their lysogens, we used comparative genomics to characterize the diversity of prophages already identified in over 10,000 prokaryotic genomes.

## RESULTS

**Prophages population and their distribution by host genome size.** To explore the presence of prophages in prokaryotic organisms (Bacteria and Archaea), we created a database of 13,713 complete genomes. These genomes had to be taxonomically assigned to at least the genus level in the NCBI database (Data Set S1 in the supplemental material). We searched for prophage signals using VirSorter (15) resulting in 33,624 prophages. Our results show that lysogens (bacteria with at least one prophage predicted) were more common (75.61%) than non-lysogens (24.38%). In addition, we found that the distribution of the genome size of prophages is clearly bimodal with two peaks at ~30 kb and ~70 kb of genome size (Fig. S1). This suggests the presence of two different populations of prophages.

Next, we wanted to determine if the presence of prophages was biased to certain bacterial taxa. For this, we first ruled out that the number of prophages was not influenced by the host genome size. Some studies have reported that the abundance of prophages positively correlates with the genome size of their host (21). Here, we found a weak positive correlation ($R^2 = 0.32$; spearman's $P$-value $< 2.2^{e-16}$) between the host genome size and the number of prophages identified. In detail, we found that genomes with sizes between 4 and 5 Mb had a higher abundance of prophages (Fig. 1A). Similar result was also observed considering only prophages of ≥30 Kb of genome length (4 and 7 Mb), and when prophages were averaged (Fig. 1B), indicating that the abundance of prophages does not only depend on the host-genome size, suggesting that the abundance of prophages in the genomes could be influenced by other factors.

**Prophages are enriched in Proteobacteria.** Surprisingly, 33,624 prophages were identified from the 10,370 genomes, with a mode and average of 1 and 3.24 prophages, respectively, and with a coefficient of variation of 60. We found that the genera *Arsenophonus* and *Plautia*, which belong to the Enterobacterales order, had some of the highest numbers of prophages, with 23 and 10 detected prophages, respectively; however, these genera only had one sequenced representative in our data set. Therefore, to avoid counts of prophages derived from a single representative, we collected all genera that were represented with at least five genomes and randomly subsampled genera that had over 50 genomes, selecting

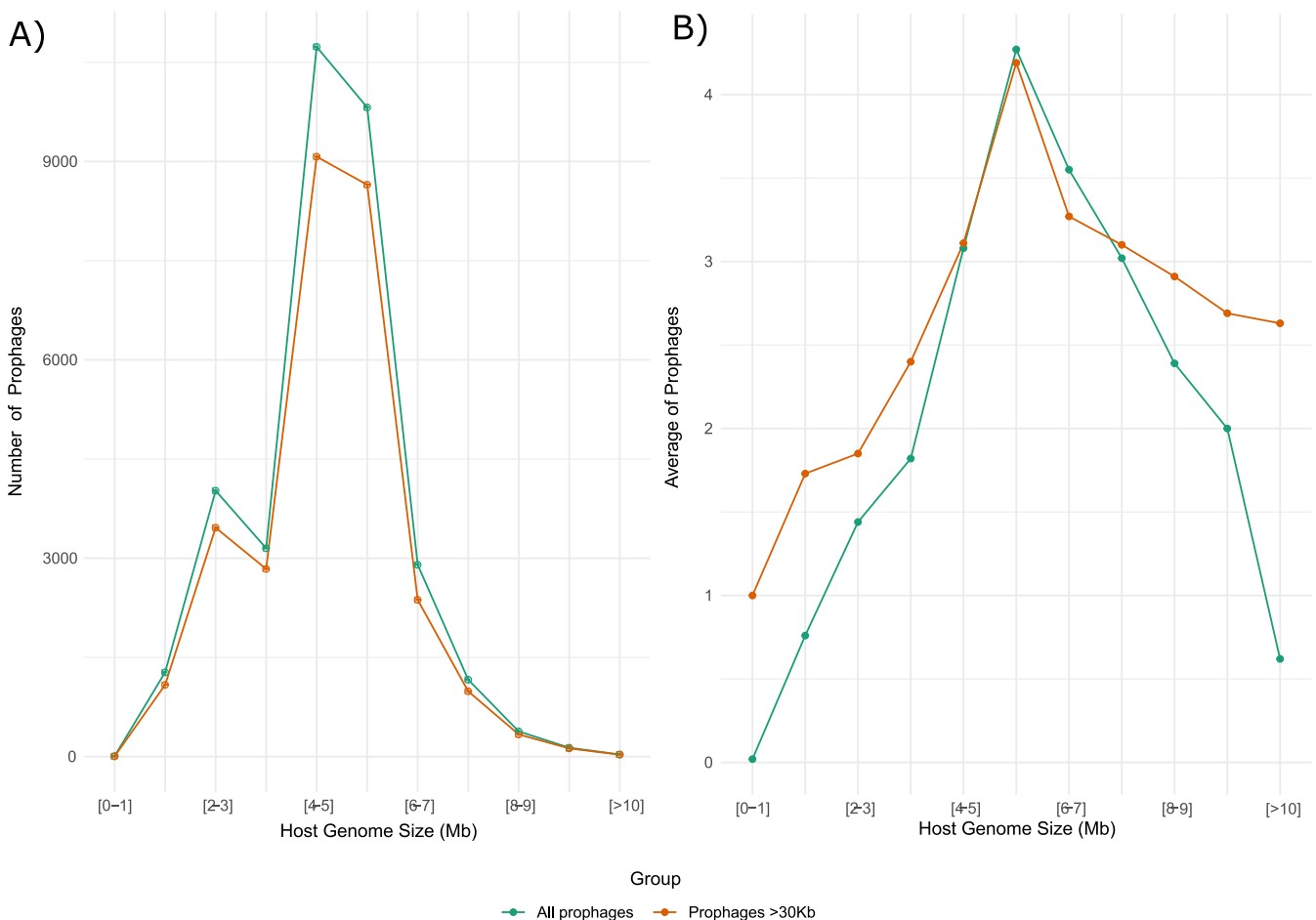

**FIG 1** Distribution of the number of prophages (A) and the average of prophages (B) per host genome based on the genome size (Mb). The total number of prophages (green line) and prophages ≥30 Kb (brown line) are shown in each plot.

only 10. We found that the members of the Proteobacteria phylum showed more prophages (Fig. 2). At the genus level, we found that *Shigella* was the genus with more prophages, with 878 prophages identified in 92 genomes and with an average prophage of 9.54, followed by *Brevibacillus*, *Escherichia*, and *Xenorhabdus* with an average prophage of 6.66, 6.43 and 6.14, respectively. Of these, the genera *Shigella* and *Escherichia* were the only ones significantly enriched in prophages with a *P*-value of $< 2.2^{e-16}$, respectively (Fig. S2 in the supplemental material). However, given that it has been reported that *Escherichia coli* and *Shigella* species are closely related and genetically constitute the same species (22, 23); here we find the same statistical significance (*P*-value of $< 2.2^{e-16}$) when we consider *Shigella* within the genus *Escherichia*. Moreover, at family level, Morganellaceae, Enterobacteriaceae, and Bacillaceae were also significantly abundant in prophages (Fig. S2). In upper taxonomy ranks, only the Enterobacterales order (five prophages on average) was significantly enriched in prophages. Thus, Proteobacteria was the phylum with more abundant prophage signals (Fig. S2), despite the that Planctomycetales and Entomoplasmatales showed abundant prophages (Fig. 2).

We further remove the Enterobacteriales order to test if any other order would appear as significantly enriched without identifying any, these results indicate that the members of the Enterobacterales are the only order significantly enriched in prophage signals (Fig. 2). In this regard (without the Enterobacterales group), we found at the genus level that the genera *Acinetobacter*, *Bacillus*, *Brevibacillus*, *Lysinibacillus*, and *Paenibacillus* were significantly enriched in prophages (Fig. S3 in the supplemental material). A possible explanation for these observations is the known bias for available genomes in public databases. In our data set, it is possible to observe a significant

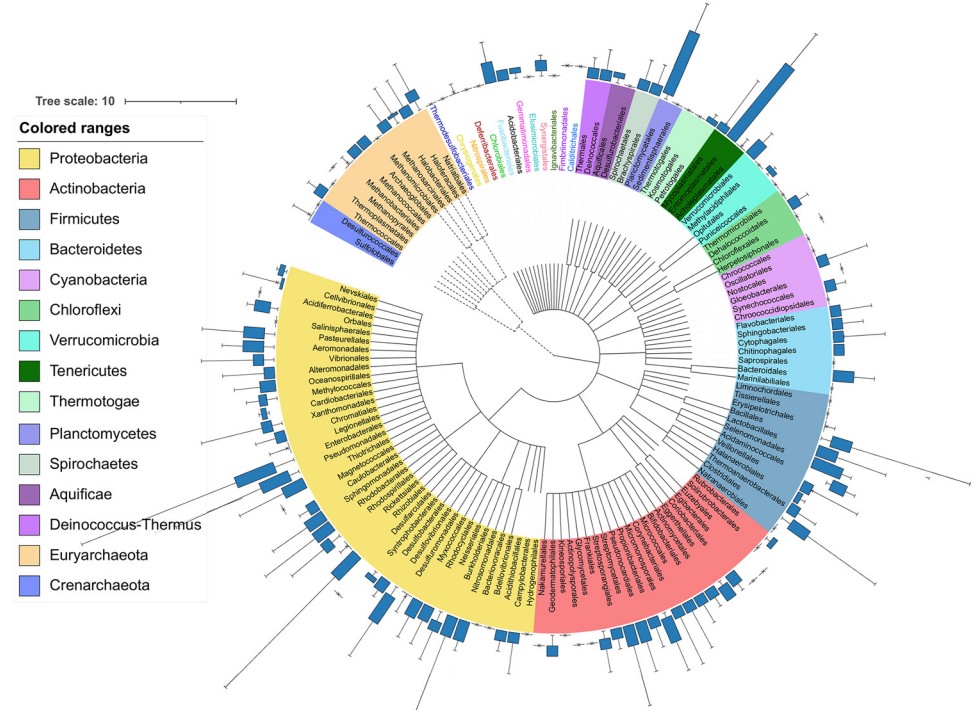

**FIG 2** Bacteria and Archaea prophage distribution. The phylogenetic tree was generated using Lifemap at order level. The phyla are shown in different colors. The Archaea and Bacteria branches are shown in dashed and solid lines, respectively.

unbalance among the available phages genomes where the most genomes belonged to Enterobacterales (2,633), Bacillales (1,419), Lactobacillales (922), and Burkholderiales (916). However, we did n ot find a significant correlation when comparing the number of prophages identified (per genera) with the number of phages reported for those genera (Fig. S4 in the supplemental material), while a significant correlation was observed between the number of prophages identified (per genera) with the number of bacterial genomes used for those genera (Fig. S4). To take into account this possible bias, we compared the abundance of prophages from 10 randomly selected genomes (see methods) and perform this step by bootstrapping (100 times) to sample all the genomes in our database. We found only nine genera enriched in prophages (Fig. 3). Of these, seven genera belonged to the Enterobacterales and two to the Bacillales. In summary, we confirm that the Enterobacterales and Bacillales harbor significantly higher numbers of prophages. In addition, several members of Proteobacteria are the only ones enriched in prophages, compared with other taxonomic groups (Fig. 2), suggesting that the abundance of prophages could be associated with their evolutionary history or with the lifestyle of their host. However, some other taxonomic affiliations showed some enrichment of prophages.

**Pathogens display high abundance in prophages.** To follow the idea, we wanted to explore further if the prophage abundance is associated within a particular taxonomic group (or groups) of the host or is influenced by other factors, such as pathogenicity. Because previous studies have reported that pathogenic strains harbor a high abundance of prophages, however, this observation was carried out in a few species, such as *Acinetobacter baumannii*, *Escherichia coli*, and *Pseudomonas aeruginosa* (11, 12, 24, 25). Here, to evaluate if the lifestyle influences the accumulation of prophages. First, we randomly collected 100 genomes for those genera that had more than 100 sequenced genomes (to avoid the bias in the number of genomes for each genus), resulting in 5,728 genomes. Of these, we selected 4,831 genomes that had associated Biosample information (see Materials and Methods). Finally, we created a data set for pathogenic hosts (*n* = 1,374 genomes with 4,623 prophages) and

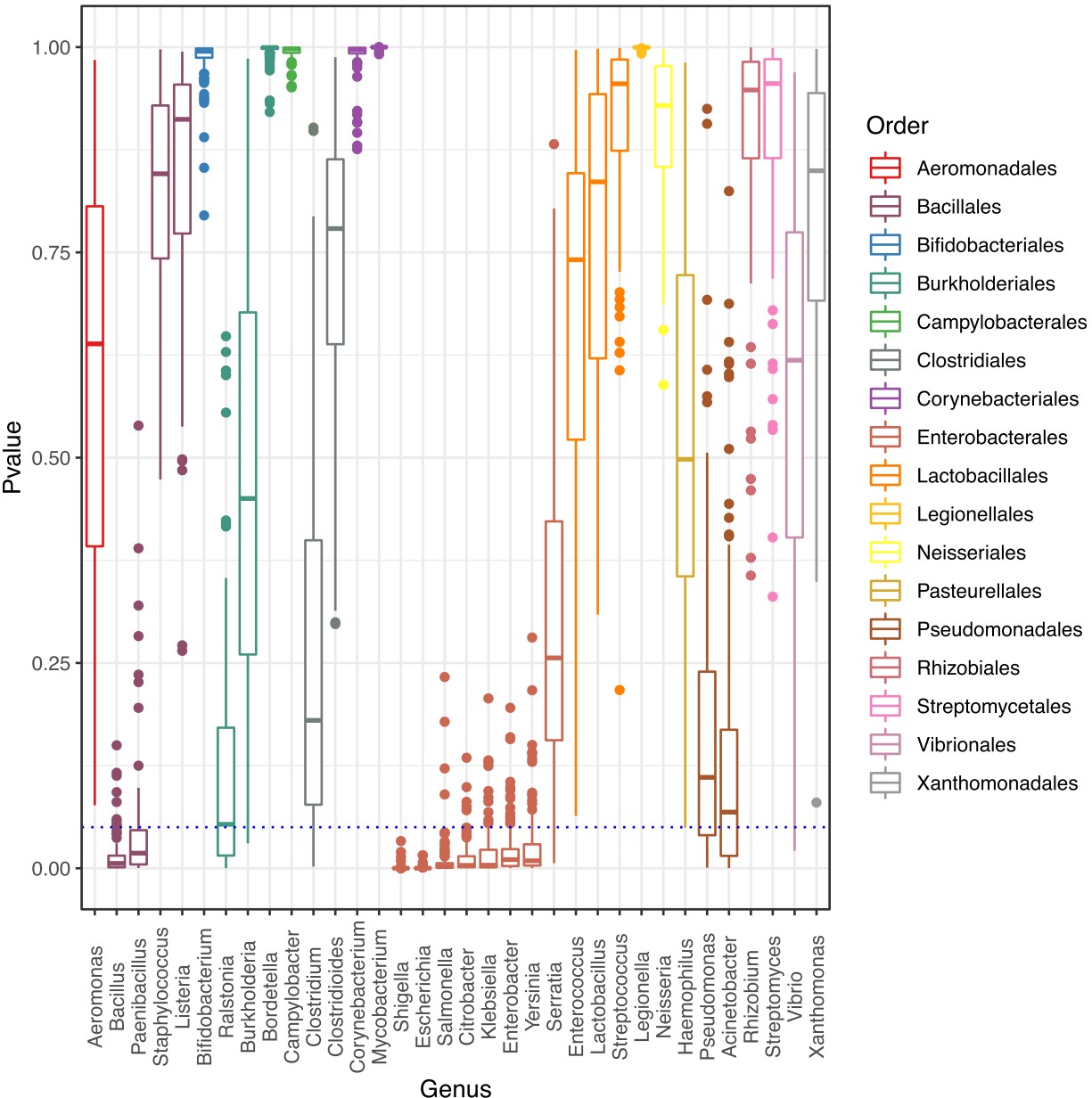

**FIG 3** Boxplots of *P*-values for different genera. *P*-values were calculated by collecting and bootstrapping (100 times randomly) the number of prophages in 10 genomes for each genus, using the Wilcoxon test (see methods). The blue dotted line indicates the significant Wilcoxon tests ($p < 0.05$). The boxplots are colored with respect to the Order taxonomic level.

another group with the rest of the genomes (unassigned group) ($n = 3,457$ genomes with 9,210 prophages). Assuming that pathogens are those which cause diseases (26), this information was collected from the Biosample of each genome (Data Set S2 in the supplemental material). However, due to the lack of information in the Biosample of several genomes, we cannot exclude the possibility that the unassigned group may contain pathogens. We found that the genomes associated with the information of a pathogenic phenotype had a mean and median of 3.36 and 3 prophages per genome, respectively, while it was 2.66 and 2 for the unassigned group. A first approximation showed a significant abundance in the group of pathogens ($P\text{-value} = 1.99^{e-15}$). However, to determine the contribution of each genus we used a Leave-One-Out (LOO) approach for this data set (see Materials and Methods) since the groups are composed of various genera (pathogenic and unassigned). We found that all genera in the pathogen group were significantly enriched (the 136 genera that make up the

**TABLE 1** Genomic host range of the viral clusters[a]

| Viral cluster (VC) Id | Present in | Host range taxonomic level | ANI between hosts |
|---|---|---|---|
| vOTU_1233 | 7 *Salmonella* and 2 *Klebsiella* genomes | Genus | 79.49% |
| vOTU_94 | 3 *Ralstonia* and 1 *Streptomyces* genomes | Phylum (Proteobacteria, Actoinobacteria) | 71.18% |
| vOTU_1254 | 2 *Odoribacter* and 1 *Parabacteroides* genomes | Family (*Odoribacteraceae*, *Tannerellaceae*) | 73.20% |
| vOTU_1158, vOTU_1237, vOTU_51, vOTU_831 | 1 *Mycobacteroides* and 1 *Mycolicibacterium* genomes | Genus | 83.11% |
| vOTU_1201 | 1 *Mixta* and 1 *Salmonella* genome | Family (*Erwiniaceae*, *Enterobacteriaceae*) | 77.11% |
| vOTU_1442 | 1 *Sodalis* and 1 *Citrobacter* genomes | Family (*Pectobacteriaceae*, *Enterobacteriaceae*) | 74.60% |

[a] The viral clusters were assigned by 90% nucleotide similarity and 80% coverage using cd-hit (see Materials and Methods). The number of prophage identified in the host genome for specific viral cluster are shown. Level of the genomic host range and the Average Nucleotide Identity (ANI) between host are displayed.

pathogen group). On the other hand, the most known bacterial pathogens have closely related environmental counterparts (11, 12, 24, 25). We compared the prophage abundance of the 109 genera shared between the pathogens and those in the unassigned group, of these, only *Enterobacter* (*P*-value of 0.008 with 268 prophages), *Acinetobacter* (*P*-value of 0.038 with 234 prophages), and *Pseudomonas* (*P*-value of 0.039 with 114 prophages) were significantly enriched in prophages in the pathogenic isolates compared with their unassigned counterparts (Table S1). Interestingly, these results are in agreement with previous reports comparing fewer genomes (11, 12, 24, 25).

**Narrow host range delimited by host phylogenetic relationship.** One of the main challenges in phage biology is to determine the host range and their genomic diversity (27). Although there are some reports of phages with a wide host range (28, 29), today there is still a debate about how wide the host range of phages actually is.

Here, to determine the genomic phage-host range, we carried out a clustering over the 33,624 prophages identified, resulting in 22,585 clustered prophages (called viral cluster [VC]). As we expected, we found that most of the prophages had a narrow genomic host range, with all the members of a given cluster associated with the same host at the genus level (88.4%). However, we found 9 VCs were composed of prophages with different host taxonomic affiliations, of which 4 VCs were identified in Proteobacteria genomes (Table 1). The most discrepant case was VC_94, whose members were found in three *Ralstonia* genomes (*Ralstonia solanacearum*) and once in a *Streptomyces* genome (*Streptomyces spongiicola*), which is relevant since these species belong to the Proteobacteria and Actinobacteria phyla (Table 1). Interestingly, members from VC_94 showed high identity (>95%) with the previously reported *Ralstonia* RSS-phages types (Fig. S5 in the supplemental material) (30–32). Following this point, we carried out a k-mer bias frequency analysis in order to identify if *Streptomyces spongiicola* is a putative specific host of RSS-phages, since viruses often share a higher similarity in k-mer patterns with their host (33, 34). We found that the k-mer frequencies of dinucleotides and trinucleotides clearly separated the RSS-phages (including the phages from the VC_94) from other *Streptomyces* phages previously reported (Fig. S6), this suggests that the prophage RSS-type sequence present in *Streptomyces spongiicola* is more likely due to contamination or a technical issue during the genome sequence and assembly than the RSS-type phage being able to infect *Streptomyces spongiicola*.

For comparison, and as a proof of the information value of k-mer frequency similarity, the VC_1254 (see Table 1), contain prophages identified in *Parabacteroides distasonis* (prophage P1097) and *Odoribacter splanchnicus* species, which showed similar k-mer frequencies suggesting that they may be related phages and although they infect different genera, both belong to the Bacteroidales order (Fig. S6 in the supplemental material**).**

**Genomic relationships between prophages.** We also wanted to determine the genomic relatedness between prophages. For this, we performed an amino acid identity (AAI) pairwise comparison analysis on the clustered prophages, since AAI was established as a reliable metric to obtain the phylogenetic relationship between phages (35, 36). Here, we retrieved 2,485 prophages that showed ≥80% of AAI with at least one other phage (where phages with ≥80% of AAI could be associated to the same genus [36]), and most of them were identified having as hosts bacteria from Proteobacteria, Actinobacteria, Bacteroides, and Firmicutes. Most of the prophages with AAI of ≥80% were prophages from hosts with

the same taxonomic affiliation (Fig. 4A). However, some Bacteroides and Actinobacteria prophages showed several connections, where a connection is defined by a pair of phages with an (AAI > 80%) with some Proteobacteria and Firmicutes prophages. Of those, the Actinobacteria prophages connected with 70 Proteobacteria and 36 Firmicutes prophages shared an average of 11.45 and 4.07 orthologous genes, respectively. In addition, we found 126 prophages with connections between Proteobacteria and Firmicutes with an average of 1.51 shared orthologous genes. Given that those genomes tend to have up to an average of 110–118 genes per genome, the fact that we observed only between 1 and 11 shared orthologs suggests more of a common pool of genes, potentially acquired by horizontal gene transfer, rather than a common evolutionary origin (Fig. 4A).

In addition, we determined the prophages relatedness by looking at the fraction of the prophages that had an AAI ≥80% associated with the same or different host-taxonomic affiliation. We found that at the genus level, the relatedness between prophages from hosts with the same taxonomic affiliation was more common (Fig. 4B). However, the relationship between prophages of different lysogens (phylogenetically distant) at the genus level was higher (0.37) and decreased as the phylogenetic relationships of the lysogens were more distant (phylum 0.0002). These results suggest that prophages could preferentially change at the genus level or come from a common gene pool, which would cause recombination events between prophages and phages (or other prophages) (37). To test this, we chose the genus *Salmonella* (all viral clusters with *Salmonella* hosts), which had the highest number of VCs, with each composed of ≥50 prophages. In total, we collected 237 *Salmonella* genomes with their corresponding prophages that came from 31 distinct VCs. We then determined the values of ANI for hosts and AAI for prophages and found that several prophages tend to cluster similarly according to their host (Fig. S7 in the supplemental material). Although the AAI values for prophages range between 20% to 100%, some clusters were homogeneous in terms of the host. For example, for the subspecies of *S. enterica* serovar Typhi, serovar Bareilly, and serovar Heidelberg, their prophages were grouped similarly to these subspecies were grouped. However, we found 20 highly similar prophages shared between multiple subspecies (Fig. S7). However, due to some clusters of prophages that are repeated on different subspecies, this result indicates that the genomic relationships of prophages are closely related to their hosts, which could explain the narrow host range in the prophages.

## DISCUSSION

To the best of our knowledge, this study comprises the most extensive analysis of prophages and describes the diversity of 33,624 prophages found in 13,713 complete prokaryotic genomes. In contrast to previous studies (21), our results indicate a weak correlation between the host genome size and prophage abundance. In this regard, Touchon and colleagues could determine a strong positive correlation between the prophage abundance and the host-genome size using 2,110 genomes (21). This correlation was observed only for genomes with a size up to 6 Mb. Moreover, they selected as bona fide those phages with a size above 30 Kb; here, we found similar results considering only prophages ≥30 Kb. However, prophages were enriched in genomes only between 4 and 7 Mb, considering all prophages and those ≥30 Kb (using a data set of 14,055 genomes). In addition, the prophage abundance was correlated with the taxonomy of the host rather than the genome size (19), indicating that temperate phage features have developed over a long phylogenetic timescale. In addition, the prevalence of the lysogens identified concurs with the prevalence of lysogens analyzed in gut metagenomics samples (19), indicating that our observations can also be obtained using different data sets. On the other hand, we observed two populations of prophages when considering the genome size. Most of the prophages found are distributed between ~30 Kb and ~60 Kb, followed by a less abundant population between ~70 Kb and ~120 Kb. These two populations could be associated with two classes of phages with respect to their genome size or their taxonomy. Some studies found that

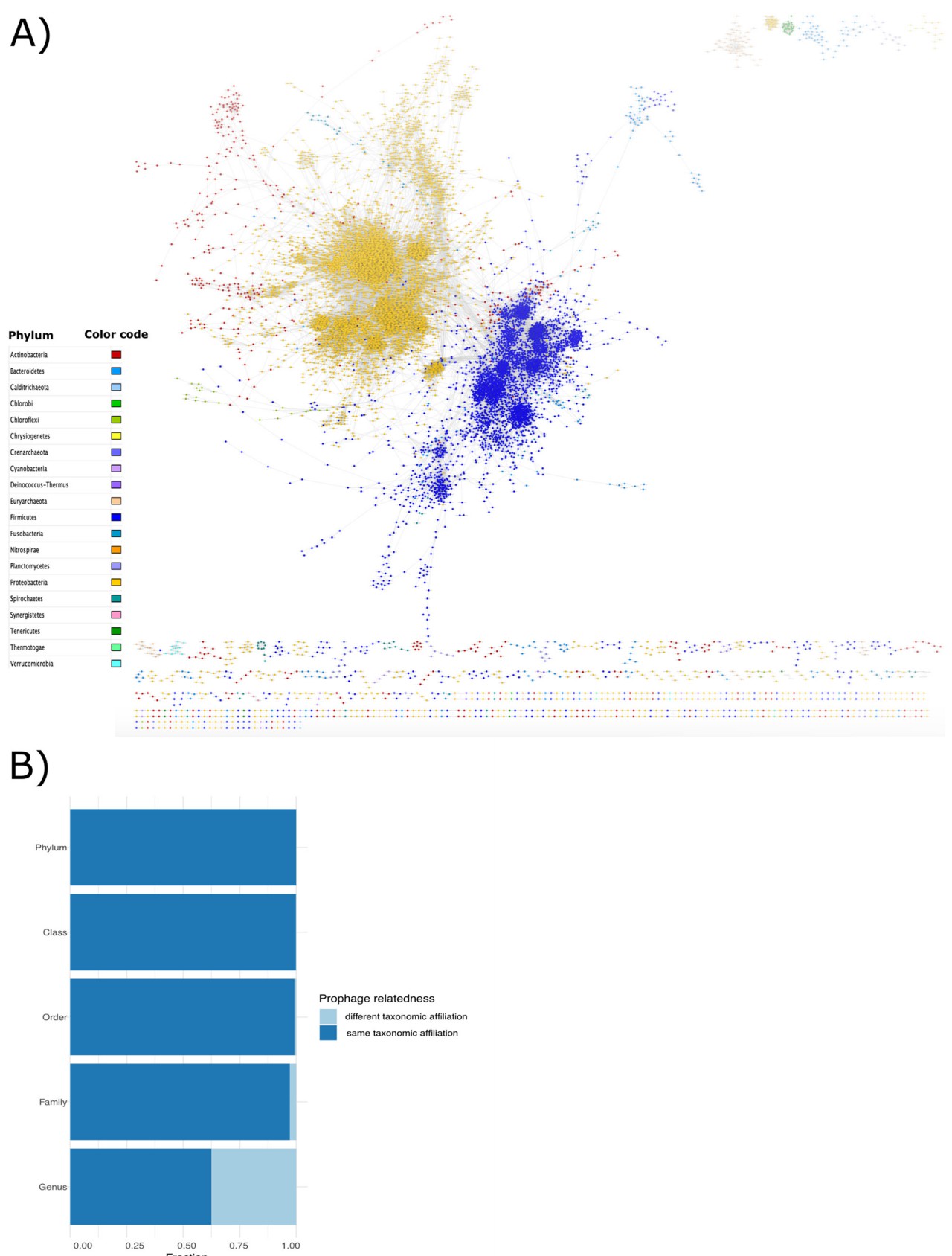

FIG 4 (A) Prophage network generated with prophages with ≥80 AAI (2,485 prophages), visualization produced with Cytoscape (see methods). Nodes represent prophages and edges represent their weighted pairwise similarities of AAI. Nodes (prophages) are depicted with different colors according to their phylum host. (B) Prophage fraction which shared an AAI value ≥80 between prophages of different lysogens at all taxonomic levels.

prophages <30Kb are degraded prophages (unfunctional), and those >30Kb correspond to the functional prophages (37).

We found that the Proteobacteria phylum, as well as the order of the Enterobacterales and most of its affiliated genera, were enriched in prophages, suggesting that the abundance of prophage signals could be associated with the taxonomy of the host (19). Additionally, by bootstrapping our data set, we confirmed that most of the collected genera from Enterobacterales were enriched in prophages as well as the Bacillales order and their affiliated genera *Bacillus* and *Paenibacillus*. However, the phage and prophage detection tools may be biased by training data and genome availability. Interestingly, most of the genera enriched in prophages have been placed at the top of the 2017 World Health Organization Priority List for Research and Discovery of New Antibiotics (38), such as *Shigella*, *Escherichia*, *Acinetobacter*, and *Pseudomonas*, because of their relevance to public health. Therefore, we consider that our results provide evidence to put more effort into studying the phage-bacteria relationships in these genera. Some reports have determined that prophages play a major role in the pathogenicity of their host by the acquisition of ARG and toxins (6, 11–13). However, this observation has been reported in Bacteria rather than Archaea. Although Archaea prophages have also been studied, Archaea pathogens have not yet been reported (39).

In this regard, we found that prophages are more abundant in bacteria associated with pathogenic phenotypes. Interestingly, these results agree with previous reports, where clinical isolates of *Acinetobacter baumannii* and *Pseudomonas aeruginosa* (24, 25) showed a higher abundance of prophages than environmental isolates. Our results indicated that the *Acinetobacter* and *Pseudomonas* genera were enriched in prophages when these species displayed pathogenic phenotypes. Therefore, prophages could play an important role in the appearance of pathogenic phenotypes for some species since, in recent years; it has been reported that prophages in *Acinetobacter*, *Pseudomonas*, *Escherichia*, and *Shigella* are involved in horizontal transfer of toxin and antibiotic resistance genes (6, 40, 41).

One characteristic of viruses is that they are highly diverse and have multiple phylogenetic origins (42). Recently, new tools have been developed to determine a good classification of viruses and to infer properly their phylogenetic relationships (36). In this study, we used two approaches (nucleotide and amino acid approaches) to determine the prophages-relationship, as previously was reported (36). As expected, most of the prophages were singletons (genome-specific) or had hosts within the same genus. In addition, some of the prophages from genetically related hosts are also related to each other. This observation was confirmed with the *Salmonella* prophages, in which both host and their prophages tend to cluster similarly. Indicating that the evolutionary relationships of the prophages are closely related to their hosts (even at the subspecies level). This has also been experimentally observed in some induced *Acinetobacter baumannii* prophages belonging to the *Vieuvirus* genus. These prophages were isolated from strains of *A. baumannii* belonging to the sequence-type 758 and only can infect strains of the same sequence-type (36). However, we found a few cases where some high similar prophages (VCs) could also be present in hosts from different taxonomic groups (Table 1), indicating that a small group of prophages may have a wide host range.

Although there are very few studies of virulent phages capable of infecting bacteria of different phyla (because virulent phages usually have a broader host range) (28, 29), there are no reports of prophages that have this characteristic. The case of the RSS-phages that we found in some genomes of *Ralstonia solanacearum* and in *Streptomyces spongiicola* HNM0071 is undoubtedly a fascinating case. However, the k-mer bias analysis and subsequent sequence analysis revealed that the RSS-phage found in *Streptomyces spongicola* HNM0071 is more likely due to some technical error than the capacity of the RSS-phage to infect the associated host.

Following this point, we wanted to determine whether temperate phages are closely related to each other (using AAI) and if highly related phages had the possibility of jumping to new hosts (infected by related phages), due to sharing a common gene pool. For this, previous studies have reported that AAI values are a good metric to

establish phylogenetic relationships between phages (36). Our results showed that most of the prophages with high similarity are limited to the evolutionary (taxonomic) scale of their hosts (Fig. 4B). Moreover, our results are consistent with other reports where they determined that lateral gene transfer between phages and their host is likely to happen principally at the genus and species level and delimited by the evolutionary relationships of their hosts (20, 29). Interestingly, these studies reported that the gene turnover occurs with high frequency among the Enterobacterales prophages (20). These observations, together with our results (where Enterobacterales are abundant in prophages), suggest that Enterobacterales could be a hot spot where prophages are the main mechanism that improves the plasticity of the genome of these species.

In addition, it is known that Proteobacteria are the most diverse phylum (43), and in the last decades, a rapid diversification of toxin genes and antibiotic resistance has been observed within the Enterobacteriaceae members (44). Therefore, future analyzes are needed to test whether phages are largely responsible for such diversification.

Finally, our findings expanded the knowledge of lysogenic interactions between prokaryotes and their prophages. We consider that these associations help to explain the relationships of prophages and their hosts in certain clades; however, more studies are needed to address the degradation processes (37) of prophages and their contribution in bacterial fitness and gene turnover.

## MATERIALS AND METHODS

**Prophage prediction.** We downloaded all the complete genomic sequences of prokaryotic organisms (13,713 genomes) reported in the reference and representative categories (45) from the RefSeq NCBI database at the end of January 2020. From these, we kept genomes with taxonomic information available at least at the genus level, to avoid potentially misclassification of genomes within the NCBI database (46, 47). To ensure reliable recovery of prophages, we removed replicons or genomes of less than 10Kb (15). Prophage prediction was carried out using VirSorter (15) with default parameters. Prophages from category 1 and category 2 (according to VirSorter) were collected. These categories represent the most reliable predictions, category 1 prophages are those prophages that contain sequences homologous to viral structural genes present in the reference database, in addition to having an enrichment in viral hallmark genes. Prophages from category 2, in contrast to category 1, are those predictions that may lack marker genes but are enriched in viral-like Caudovirales genes or distinct viral genes. In addition, we validated these predictions using VIRALVERIFY (48) by removing all prophage predictions that were tagged as plasmids. The resulting prophages were considered as bona fide prophages and were kept for downstream analysis. Then, to evaluate prophage abundance enrichment at the different taxonomic levels (host), we coupled a Wilcoxon test with a bootstrapping approach for genera with at least 50 genomes. In brief, for the bootstrapping, we collected 10 random genomes for each genus at each iteration. This step was repeated 100 times (bootstrapping).

**LOO approach.** The Biosample information was obtained from the genomes listed in Data set S1 using efetch form E-utilities (49). We collected the metadata of the following sections: general description, isolation source, isolation site, host, environmental medium, and sample type. Then, to consider whether a genome corresponds to a pathogenic phenotype, we manually checked the information of each section. If they were listed as "pathogen" or "pathogenic" in any of the aforementioned fields or if the isolation source was from a patient, animal, or plant associated with a disease, they were considered pathogens. Otherwise, they were considered as unassigned. Next, to determine the contribution of the prophages from each genus in the enrichment between pathogens and the unassigned group, we took out one genus at a time from both groups and analyzed the respective enrichment, termed the LOO approach. Statistical test (Wilcoxon test) was carried out with the wilcox.test function implemented in $R$. We considered an adjusted $P$-value less than or equal to 0.05 as an indicator of statistical significance. The $P$-value correction was performed by the p.adjust function in R using the Bonferroni method.

**Prophage clustering.** All the prophages with at least 90% nucleotide similarity over at least 80% of the genome length were clustered using cd-hit (50). Furthermore, an analysis of Average Amino acid Identity (AAI) was carried out based on a pairwise comparison of the dereplicated prophage sequences (clustered prophages) using CompareM (https://github.com/dparks1134/CompareM). Then, we calculated the average AAI overall putative homologs, an AAI $\geq$ 80 was used to consider significant relatedness among prophages at the genus level, as previously reported (35, 36). The prophage relatedness was visualized as an AAI network using Cytoscape v.3.8 (51).

**K-mer usage bias.** To identify the host-phage relationship we used the co-occurring patterns of k-mer frequency between prophages and their host genome. Since viruses often share higher similarity in k-mer usage with its host we used k-mer usage bias metric as a proxy to quantify how similar k-mer usage profiles were between the prophage and bacterial genomes (33, 34). For this, the k-mer bias measures were obtained for selected bacterial and phages genomes using the k-mer sizes of 1, 2, 3, 4. The calculation of the k-mer usage bias was performed according to the mathematical formula proposed

in previous studies (52). Briefly, the observed frequency of a k-mer is normalized by the expected frequency, which is calculated as the product of all the frequencies of the subkmers from that k-mer.

**Tree Of Life (TOL).** The TOL at order level for Archaea and Bacteria was generated using Lifemap (53). The TOL was annotated in iTOL (54).

## SUPPLEMENTAL MATERIAL

Supplemental material is available online only.

**DATA SET S1**, XLSX file, 0.4 MB.
**DATA SET S2**, XLSX file, 0.3 MB.
**FIG S1**, PDF file, 0.01 MB.
**FIG S2**, PDF file, 0.7 MB.
**FIG S3**, PDF file, 0.1 MB.
**FIG S4**, PDF file, 0.03 MB.
**FIG S5**, PDF file, 0.1 MB.
**FIG S6**, PDF file, 0.1 MB.
**FIG S7**, PDF file, 0.5 MB.
**TABLE S1**, DOCX file, 0.1 MB.

## ACKNOWLEDGMENTS

G.L.L. received a postdoctoral fellowship (2019-000012-01EXTV-00488) from CONACyT. We are thankful to Alfredo Hernández-Alvarez and Víctor Del Moral-Chávez for technical support. G.L.L. thanks to Juan Sebastian Andrade Martinez, Luis Alberto Chica Cárdenas, Laura Milena Forero Junco, Ruth Hernandez Reyes, Leonardo Moreno Gallego, and Laura Avellaneda Franco, members of the Viromics group within the Max Planck Tandem Group in Computational Biology, for the general discussion of the results presented in this manuscript. G.L.L. special thanks to Guillermo Rangel for his comments on the manuscript.

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
