## [Reviewer comments · mSystems]

Mining of thousands of prokaryotic genomes reveals high abundance of prophages with a strictly narrow host range.

Gamaliel López-Leal, Laura Camelo-Valera, Juan Hurtado-Ramírez, Jérôme Verleyen, Santiago Castillo-Ramírez, and Alejandro Reyes-Muñoz

Corresponding Author(s): Gamaliel López-Leal, Universidad de Los Andes

Review Timeline:

Submission Date:	April 7, 2022
Editorial Decision:	April 25, 2022
Revision Received:	May 4, 2022
Accepted:	June 20, 2022

Editor: Rup Lal

Reviewer(s): The reviewers have opted to remain anonymous.

Transaction Report:

DOI: <https://doi.org/10.1128/msystems.00326-22>

April 25, 2022

Dr. Gamaliel López-Leal
Universidad de Los Andes
Departamento de Ciencias Biológicas
Bogotá, D.C. 04510
Colombia

Re: mSystems00326-22 (Mining of thousands of prokaryotic genomes reveals high abundance of prophages with a strictly narrow host range.)

Dear Dr. Gamaliel López-Leal:

Thank you for submitting your manuscript to mSystems. We have completed our review and I am pleased to inform you that, in principle, we expect to accept it for publication in mSystems. However, acceptance will not be final until you have adequately addressed the reviewer comments.

Preparing Revision Guidelines

Sincerely,

Rup Lal

Editor, mSystems

Journals Department
Reviewer comments:

Reviewer #1 (Comments for the Author):

This paper addresses an interesting subject, which is the distribution of prophages in prokaryotic genomes and the relationships between them and their hosts. In addition to my previous review, I notice that the authors addressed the modifications suggested by the reviewers, which noticeably improved the paper.

There are a few typo mistakes:

On page #2, line #35, the word ". Make" must be replaced by ", making".

On page #13, line #276, the word "the" must be "they".

On page #13, line #295, a comma is needed between the word "genome" and the word "the".

On page #14, line #313, the word ". Some" must be replaced by ", some".

On page #15, line #334, the word ". Indicating" must be replaced by ", indicating".

Reviewer #2 (Comments for the Author):

In this study, López-Leal and colleagues detected and analyzed prophages across a large dataset of bacterial genomes. They found that prophages appear more prevalent in Enterobacterales and in pathogenic strains. Overall, this is a good study, but most of the analyses are rather descriptive in nature. I have several comments that need to be addressed:

The authors analyzed only 13,713 genomes on RefSeq. This number seems very low, were some other criteria used to filter the dataset besides completeness?

The Methods section is not very detailed, and more clarifications are needed. In particular, the criteria used to detect prophages are not very detailed. For instance, what are categories 1 and 2 of VIRsorter? The k-mer analysis is also not very detailed: more information should be added. What equation was used?

Shigella and Escherichia coli are treated as different clades but they are actually the same species. Several studies have found that the diverse Shigella 'species' are in fact different strains of E. coli.

Could there be a detection bias towards enterobacteria? This should be discussed. Many model species are enterobacteria and the knowledge of phages is also much richer for this clade.

How many shared genes were used for the AAI analysis? It is not very informative to use AAI when very few genes are shared between genome pairs.

Salmonella analysis: what if prophages are vertically inherited? Or if the same strain is sequenced multiple times? This is likely to impact the authors' results.

I don't really see how the results of this study contradict the results from Touchon et al. ? The results seem largely congruent from what I can see but maybe I am missing something.

Reviewer #1 (Comments for the Author):

This paper addresses an interesting subject, which is the distribution of prophages in prokaryotic genomes and the relationships between them and their hosts. In addition to my previous review, I notice that the authors addressed the modifications suggested by the reviewers, which noticeably improved the paper.

R: We thank the reviewer for taking the time to review our manuscript.

There are a few typo mistakes:

On page #2, line #35, the word ". Make" must be replaced by ", making".

R: This has been fixed (see line 35).

On page #13, line #276, the word "the" must be "they".

R: This has been fixed (see line 283).

On page #13, line #295, a comma is needed between the word "genome" and the word "the".

R: This has been fixed (see line 302).

On page #14, line #313, the word ". Some" must be replaced by ", some".

R: This has been fixed (see line 320).

On page #15, line #334, the word ". Indicating" must be replaced by ", indicating".

R: This has been fixed (see line 341).

Reviewer #2 (Comments for the Author):

In this study, López-Leal and colleagues detected and analyzed prophages across a large dataset of bacterial genomes. They found that prophages appear more prevalent in Enterobacterales and in pathogenic strains. Overall, this is good study, but most of the analyzes are rather descriptive in nature. I have several comments that need to be addressed:

R: We thank the reviewer for her/his comments and for taking the time to review our manuscript. We have taken into account her/his suggestions in the revised version of the manuscript.

The authors analyzed only 13,713 genomes on RefSeq. This number seems very low, were some other criteria used to filter the dataset besides completeness?

R: Thank you for this observation. We use all RefSeq complete prokaryotic genomes within the "Reference" and "Representative" genome categories computationally curated based on assembly quality and annotation measures. Please take into consideration that RefSeq is a curated reference set of sequences derived from all available genomes in GenBank, so it is expected to have significantly less (but highly curated) data compared to GenBank. To date, there are reported 15,497 genomes at the NCBI (<https://www.ncbi.nlm.nih.gov/genome/browse#!/prokaryotes/>). Additionally, at least at the genus level, we kept all those genomes with reliable taxonomic information available. We

consider that 13,713 genomes are a considerable number of genomes if we take into account that these genomes cover a wide diversity of bacterial species. Moreover, the total number of prophage signals was 33,624, which is considerably bigger and is 1.45 fold greater than the 13,713 genomes on the RefSeq. We have added some of this information in lines 87 and 153.

The Methods section is not very detailed, and more clarifications are needed. In particular, the criteria used to detect prophages are not very detailed. For instance, what are categories 1 and 2 of VIRsorter? The k-mer analysis is also not very detailed: more information should be added. What equation was used?

R: Thank you for these suggestions. To clarify, in VirSorter category 1 is the most confident prophage prediction, which contains several marker genes (main criterion) plus one or more characteristics of the secondary criteria, based on enrichment in hypothetical or uncharacterized genes, viral genes, gene size, gene orientation. In other words, the prophages grouped in category 1 contain sequences homologous to phage structural genes present in their custom database and have enrichment in viral genes ("most confident" predictions). In contrast, in category 2 the predictions may not have marker genes but are enriched in viral genes (from the caudovirales dataset), or a distinct viral gene has been detected or associated with at least one other metric (secondary criterion). Generally, prophages grouped in this category are considered "likely" predictions (see lines 93-99). Considering the k-mer analysis, the k-mer frequency was normalized by the expected frequency, which is calculated as the product of all the frequencies of the sub-k-mers that form the k-mer. We have added this information in lines 134-143.

Shigella and *Escherichia coli* are treated as different clades but they are actually the same species. Several studies have found that the diverse *Shigella* 'species' are in fact different strains of *E. coli*.

R: Thank you for this observation. We also find enrichment in prophages when considering *Shigella* as part of the genus *Escherichia coli* (p -value of $< 2.2e-16$). We have added some lines describing this result (see lines 187-190).

Could there be a detection bias towards enterobacteria? This should be discussed. Many model species are enterobacteria and the knowledge of phages is also much richer for this clade.

R: Thanks for the comment. According to the NCBI viral database (which uses VirSorter to predict phage regions), the most extensive collection of phages corresponds to *Mycobacterium* (1,933 phages), *Streptococcus* (1,083 phages), and *Escherichia* (929 phages), which belong to the orders Actinomycetales, Lactobacillales and Enterobacterales, respectively. We found only the order Enterobacterales to be enriched in phages of these orders. However, we consider that phage and prophage detection tools may be biased by training data and the availability of viral genomes. We have discussed this issue in lines 202-205 and 357-359.

How many shared genes were used for the AAI analysis? It is not very informative to use AAI when very few genes are shared between genome pairs.

R: The CompareM tool uses the complete genomes of the prophages to perform the gene calling, find the orthologous genes between them and thus determine the AAI value. Therefore, the number of orthologous genes varies between each prophage comparison. We have highlighted in the manuscript that the most distant prophages share orthologous genes (with an average of 1-11 orthologs genes). This information is on lines 297-304.

Salmonella analysis: what if prophages are vertically inherited? Or if the same strain is sequenced multiple times? This is likely to impact the authors' results.

R: Although it is known that prophages can be transmitted vertically by integrating into the host genome, it has also been reported that prophages usually undergo a degradation process once inside the host genome. In this sense, only a few cases could explain our observations. However, more analysis is needed to elucidate this point. In addition, in the case of *Salmonella*, the genomes used (according to NCBI metadata) come from bacteria isolated in different years and locations around the world. Importantly, although there are strains sequenced several times, our dataset does not contain redundant genomes because we selected representative (non-redundant) genomes from the NCBI database (see above).

I don't really see how the results of this study contradict the results from Touchon et al. ? The results seem largely congruent from what I can see but maybe I am missing something.

R: Thanks for the comment. When considering prophages >30Kb our results and Touchon's agree rather well yet there are some important differences when the whole range of prophage and bacterial genomes sizes are considered. We add this statement to make it clearer to the reader (see lines 336-340).

June 20, 2022

Dr. Gamaliel López-Leal
Universidad de Los Andes
Departamento de Ciencias Biológicas
Bogotá, D.C. 04510
Colombia

Re: mSystems00326-22R1 (Mining of thousands of prokaryotic genomes reveals high abundance of prophages with a strictly narrow host range.)

Dear Dr. Gamaliel López-Leal:

Your manuscript has been accepted, and I am forwarding it to the ASM Journals Department for publication. For your reference, ASM Journals' address is given below. Before it can be scheduled for publication, your manuscript will be checked by the mSystems production staff to make sure that all elements meet the technical requirements for publication. They will contact you if anything needs to be revised before copyediting and production can begin. Otherwise, you will be notified when your proofs are ready to be viewed.

Publication Fees:

We recognize that the video files can become quite large, and so to avoid quality loss ASM suggests sending the video file via <https://www.wetransfer.com/>. When you have a final version of the video and the still ready to share, please send it to mSystems staff at mSystems@asmusa.org.

For mSystems research articles, if you would like to submit an image for consideration as the Featured Image for an issue, please contact mSystems staff at mSystems@asmusa.org.

Sincerely,

Rup Lal
Editor, mSystems

Journals Department
Supplemental Material 6: Accept
Supplemental Material 10: Accept
Supplemental Material 1: Accept
Supplemental Material 9: Accept
Supplemental Material 4: Accept
Supplemental Material 3: Accept
Supplemental Material 5: Accept
Supplemental Material: Accept
Supplemental Material 2: Accept
Supplemental Material 8: Accept